# SWEb: A Large Web Dataset for the Scandinavian Languages

**Tobias Norlund,**[*] **Tim Isbister, Amaru Cuba Gyllensten, Paul Dos Santos, Danila Petrelli, Ariel Ekgren, Magnus Sahlgren**
AI Sweden

## Abstract

This paper presents the hitherto largest pretraining dataset for the Scandinavian languages: the Scandinavian WEb (SWEb), comprising over one trillion tokens. The paper details the collection and processing pipeline, and introduces a novel model-based text extractor that significantly reduces complexity in comparison with rule-based approaches. We also introduce a new cloze-style benchmark for evaluating language models in Swedish, and use this test to compare models trained on the SWEb data to models trained on FineWeb, with competitive results. All data, models and code are shared openly.

## 1 Introduction

Large language models have made significant strides in recent years due to their general capabilities in language-processing tasks. This progress has been largely driven by the development of extensive and high-quality pretraining datasets sourced from open web data (Wenzek et al., 2020; Brown et al., 2020; Abadji et al., 2022; Penedo et al., 2023; 2024). However, the majority of research aimed at improving pretraining data focuses on high-resource languages such as English. Our goal is to create a large-scale and high-performing open pretraining dataset specifically for the Scandinavian (*north-germanic*) languages: Swedish, Danish, Norwegian, and Icelandic.

Existing large-scale datasets for these languages primarily include mC4 (Xue et al., 2021), OSCAR (Abadji et al., 2022), and HPLT Datasets 1.2 (de Gibert et al., 2024). The Scandinavian portion of mC4 comprises approximately 100B tokens, 10B tokens for OSCAR 23.01, and 35B tokens for HPLT, which are all relatively small numbers considering that state-of-the-art large language models today are trained on trillions of high-quality tokens.

In this paper we make the following contributions:

- We release[1] the largest to date pretraining dataset for the Scandinavian languages: **S**candinavian **WEb** (**SWEb**). SWEb is the result of running our proposed pipeline on 98 Common Crawl snapshots. SWEb contains 1.01 trillion tokens in the Scandinavian languages, approximately an order of magnitude more than other available open alternatives.

- We introduce a new cloze-style benchmark for evaluating language models in Swedish, **HP-MEK**, a subset of the Swedish Scholastic Aptitude Test (Högskoleprovet) used for university admissions in Sweden. Using HP-MEK, we show our data performs on-par with data from the recently proposed FineWeb (Penedo et al., 2024) pipeline.

- We propose a new comprehensive pipeline for curating pretraining data for large language models, built around a model-based text extractor that significantly reduces complexity and is easily adaptable through rapid data annotation[2]. Most notably, we demonstrate that our pipeline returns about +60% more high quality tokens than FineWeb on the same input data.

---

[*]Corresponding author: `tobias.norlund@ai.se`
[1]Data available here: `https://huggingface.co/datasets/AI-Sweden-Models/SWEb`
[2]Code and extractor model is available here: `https://github.com/aidotse/SWEb`

## 2 BACKGROUND AND RELATED WORK

Early efforts to extract massive amounts of text from the open internet for LLM training start from WebText (Radford et al., 2019), developed for training GPT-2. In this case, outbound links from Reddit with a certain number of upvotes were used as the content selection criterion. Text was extracted using Dragnet (Peters et al., 2018) and Newspaper[3] and filtered with several heuristics, resulting in a dataset of 40GB after deduplication. Soon after, CCNet (Wenzek et al., 2020) and C4 (Roberts et al., 2019) were proposed, both based on open web data from Common Crawl. C4 was initially developed exclusively for English but was later followed by a multilingual version, mC4 (Xue et al., 2021). CCNet, on the other hand, was multilingual from the outset.

Both CCNet and C4 are based on the WET archives from Common Crawl, where all HTML formatting has been stripped, leaving only the text. However, this text still contains a significant amount of noise in the form of menu and ad text, headers, footers, and sidebars, which are irrelevant to the page's primary content. A successful method for extracting primary content from WET archives is to deduplicate the documents at the line level. C4 globally deduplicates all lines, while CCNet deduplicates over a subset of documents from the same Common Crawl dump. Line-by-line deduplication is the primary extraction method in CCNet, whereas C4 additionally employs a range of English-specific cleaning heuristics.

Following extraction comes a language detection and filtering step. Whilst more computationally expensive, performing language detection post extraction been shown to achieve better detection accuracy than filtering pre extraction (especially for low-resource languages) (Wenzek et al., 2020). Quality filtering differs slightly between the two, with C4 filtering using several heuristics, a bad words filter, and URL deduplication. In contrast, CCNet employs a model-based filter, using perplexity as a quality measure with a KenLM model trained on Wikipedia.

CCNet has since been utilized in subsequent works such as RedPajama (v1 and v2) (Together Computer, 2023) and Dolma (Soldaini et al., 2024). RedPajama-Data v2 runs CCNet on an expanded number of Common Crawl snapshots and filters for five high-resource languages (none of which are Scandinavian, however). They also extend CCNet's quality filtering by pre-computing a larger set of popular quality signals but leave the thresholding and filtering to the user.

Recently, several works have moved away from Common Crawl's WET archives in favor of processing the raw HTML of webpages found in the WARC archives. Utilizing mor sophisticated text extraction turns out to be critical for the improving quality of the resulting data (Penedo et al., 2024). In MassiveWeb (Rae et al., 2021), the tree structure of HTML is utilized to more easily group and identify the primary content of pages. Some formatting is also retained, with the argument that this "diversity in formatting style translates effectively to the generative capabilities of the Gopher models."

A similar approach is developed in NeuScraper (Xu et al., 2024), where a model is trained to – on an element level – decide whether it should be extracted or not. Both RefinedWeb and FineWeb use the open-source framework Trafilatura (Barbaresi, 2021) to extract text from HTML. Trafilatura is based on rules and heuristics on the DOM tree to identify primary content and has been shown to be the best non-commercial extractor for certain domains (Lopuhin, 2019). However, quality issues are still prevalent, and in RefinedWeb (Penedo et al., 2023) further (line-based) filters are added in an attempt to address these.

MassiveWeb introduce what they call "repetition filters" to remove documents with repetitive text, that is found beneficial with their extractor. These are also sucessfully reused in both RefinedWeb and later FineWeb. Through a systematic analysis, FineWeb further adds a small set of quality filters, that is shown through ablation experiments to yet increase quality. For a state of the art pipeline like FineWeb, the filtering can add up to 30 or more quantities and rules that might be difficult to oversee and adapt to new languages.

---

[3]https://github.com/codelucas/newspaper

Figure 1: The SWEb pipeline. We use Common Crawl's preprocessed WET archives for content selection, and WARC for extraction. At the center stage sits our model based Markdown extractor, that is the primary workhorse to produce our dataset.

# 3   THE SWEB PIPELINE

As evident by the previous section, much focus has been placed on the development of heuristics and filters to enhance the quality of the resulting data.

To move away from the extensive number of manual thresholds and complex extraction rules, we propose a more data-driven alternative. By learning a model for extraction, this complexity can be significantly reduced.

We begin by describing our pipeline that, like existing approaches, consists of the overarching steps of content selection, extraction, quality filtering, and deduplication (Figure 1).

## 3.1   STAGE 1: CONTENT SELECTION

Our pipeline begins with content selection, which aims to identify and select source documents from Common Crawl that are likely to be in one of the Scandinavian languages. Since the Scandinavian languages make up a very small portion of the entire Common Crawl, we want to implement this step early to filter out all non-relevant content.

We use CCNet to identify Scandinavian documents within the entire Common Crawl dataset. CCNet processes the WET archives, and after line-by-line deduplication, language detection is performed using fastText (Joulin et al., 2016b). Documents with a detected language score above 0.2 for any of the four languages are selected for the next stage.

## 3.2   STAGE 2: CONTENT EXTRACTION AND FORMATTING

In Stage 2, we start from the documents indentified in Stage 1 but discard their content and instead use Common Crawl's index to download their original HTML from the WARC archives. This means we use CCNet and the WET documents solely for content selection, but not for extraction. In the WET archives, all formatting and structure, such as header information, tables, text styles, bullet lists, and images, have been removed. We believe it is useful for language models to also model such structural information, in addition to plain text. Therefore, we aim to extract also this information from the webpages, and retain it in Markdown format.

We propose a new method for extracting primary content from the webpages, consisting of two steps: 1) Convert HTML to Markdown, 2) Extract primary content from the resulting Markdown through line-by-line filtering with a trained model.

### 3.2.1   CONVERT HTML TO MARKDOWN

Since we want to preserve basic textual formatting, we choose to convert from HTML to Markdown with its very lightweight markup, thus does not add many extra tokens. We convert all incoming HTML documents to Markdown using Pandoc, stripping links and images. See Listing 1 for an example.

No extraction has yet taken place, so these documents are still full of noise from menus, advertisements, and other extraneous content. We address this in the next step.

Listing 1: A webpage converted to markdown (translated, originally in Swedish), including title, top menu, headings and primary content. The document is truncated for brevity.

```
1  My Life, My Thoughts & My Training
2
3  ## The Blog
4  - The Blog
5  - Running Times Over the Years
6  - My Education
7  - Personal Training
8
9  ## Wednesday, December 14, 2011
10
11 ### The Tough Week Continues...
12
13 ...but tomorrow is a rest day.
14
15 I can feel in my body that I am right in the middle of a tough week *(I periodize my training, among other
       things, by alternating between heavy, medium, and light weeks.)* and running was not exactly the first
       thing I thought about when I woke up this morning. But after a nap together, sleep?\!
16
17 Posted by
18
19 Running & Life at
20 ...
```

### 3.2.2 MODEL-BASED CONTENT EXTRACTION

We observe that individual lines in the Markdown documents often correspond to specific elements such as headers, paragraphs, or navigation links. This makes lines an appropriate level for extraction. Therefore, we develop a custom annotation tool (details in Appendix B) to annotate which lines in these Markdown documents should be extracted and which should not. We ask annotators to mark what is considered the "main content" on the current webpage, and make some principled decisions for quality and consistency:

1. We do not extract navigation text such as menus or buttons.
2. A significant portion of the webpages are product pages. We decide to extract these only if there is a product description consisting of at least three complete sentences.
3. We extract tables if they are well-formatted and their content is tightly coupled to the main content.
4. On blogs or article pages that include user comments, we extract such comments in addition to the main content.
5. We do not extract information from sidebars unless it clearly constitutes main content.

While not explicitly excluded as per our guidelines, advertisement text isn't considered to be main content and is thus implicitly excluded. The full annotation guidelines can be found in Appendix C. In total, we annotate 1,380 webpages, using 100 of these for validation and the remainder as training data for our extraction model.

**Line Extraction Model**   Our dataset consists of Markdown documents with corresponding binary line annotations, see Figure 11. We aim to train a model to predict this label for each line. For this purpose, we choose to use a transformer encoder, where each newline is replaced with a special token [SEP]. We then feed the entire document through the encoder, with each [SEP] token representing the preceding line. This way, each line classification is contextualized by (theoretically) the full document context.

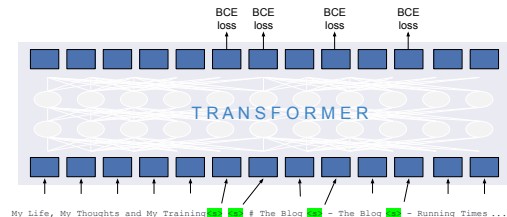

Figure 2: Illustration of our proposed line classification model. Each newline is replaced by a special  token, and the corresponding embeddings are used for classification

$$h_{0:n} = \text{Encoder}(x_{0:n}) \qquad (1)$$

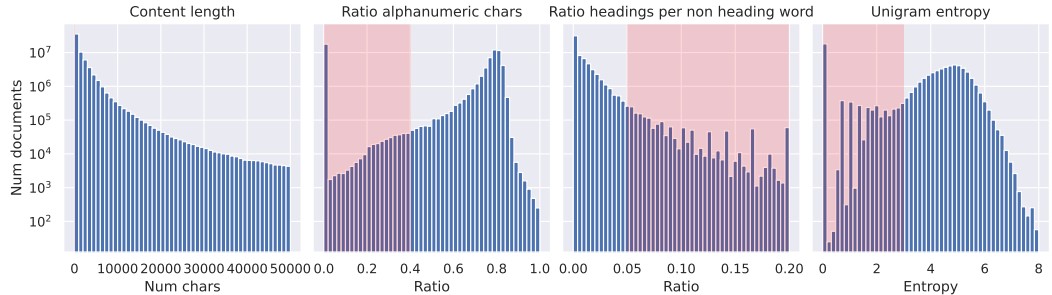

Figure 4: Filtering distributions on two Common Crawl dumps, and exclude regions marked in red. We exclude documents whose content length is shorter than 100 chars (invisible in the chart).

Through a linear projection of the output hidden state of each [SEP] token, we obtain logits for predicting the binary label of the current line. Let $j$ denote the token index corresponding to each [SEP] token in the document. We then get the predicted probability for the line as:

$$p_j = \sigma(Wh_j + b) \tag{2}$$

where $\sigma$ is the sigmoid function. The model is trained using binary cross-entropy loss between each $p_j$ and an annotated line label. See Figure 2 for an illustration. We apply a fixed threshold to $p_j$ to determine whether to include or exclude the line.

The Markdown documents can be very long, so we use the Longformer (Beltagy et al., 2020) architecture. Specifically, we use a pre-trained model that supports up to 16k tokens and has been trained for representation learning using masked language modeling[4]. The Longformer is a linear complexity transformer, thanks to its local self-attention, where each token only attends to a fixed number of nearby tokens. We use a local attention window size of 256 tokens and no global attention, as this turned out to only impair generalization.

We fine-tune the entire model on our training set of 1,280 documents, and the results on the validation set can be seen in Figure 3. We use the Adam optimizer with a constant learning rate of 1e-5. The results show that despite our small-scale training data, we achieve an F1 score of 87%. Some more details and error analysis is provided in Appendix D.

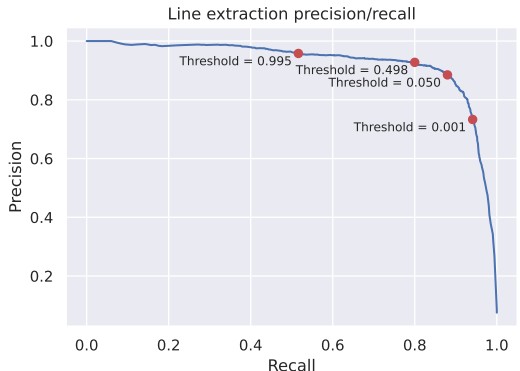

Figure 3: Precision/recall of our final line extraction model. We pick a threshold of 0.05 at inference, e.g. when applying the model for extraction.

Finally, we normalize the text using Fix Text For You (Speer, 2019).

### 3.3 STAGE 3: QUALITY FILTERING AND CLEANING

The third stage aims to filter for quality, reduce duplicate content and remove personally identifiable information (PII).

**Quality Filtering** A significant advantage of our model-based extraction is that it also implicitly performs much of the quality filtering. The extractor effectively learns to exclude content that is not of sufficient quality, such as spam and advertisements. This allows us to use only a minimal set

---

[4]https://huggingface.co/severinsimmler/xlm-roberta-longformer-base-16384

of simple filters to remove edge cases where the extractor fails. Through qualitative analysis, we developed four filters to exclude such edge cases:

1. **Content length**: We exclude cleaned documents that are shorter than 100 characters.

2. **Ratio of alphanumeric characters**: We exclude cleaned documents whose ratio of alphanumeric characters is lower than 0.4. These documents primarily consist of data tables and are not relevant without additional context.

3. **Headings per non-heading word**: We note that in some documents, only headings are extracted with little or no accompanying text. We compute the ratio of the number of headings to the total number of words from non-heading lines. If the ratio is greater than 0.05, we exclude the document.

4. **Unigram entropy**: Also used in Together Computer (2023), this measures the diversity of the content and is computed using $\sum -x/total * \log(x/total)$ where the sum is taken over counts of unique words in the normalised content. By manual inspection, we found a threshold value of 3.0 to be reasonable, and exclude all documents below it.

In Figure 4, we show the distributions of these four quantities, and in Appendix E, we provide examples of documents that are filtered out.

**De-duplication** We used MinHashLSH (Leskovec et al., 2020) for document level near duplicate removal. The MinHash signatures were computed using unicode code point-level shingles of size 16[5], 14 bands, and 8 hashes per band (a total of 112 Hashes). Deduplication was done per band in an iterative fashion: For each band in order, we grouped documents by their hashes within that band, and kept only one document per group. Following FineWeb (Penedo et al., 2024), we only performed deduplication within snapshots, and not between them, as this was shown to increase downstream performance.

**PII Replacement** As a final processing step, we make a best effort at removing personally identifiable information from our data. To this end, we use regular expressions to replace email addresses and publicly facing IP-adresses with one of a few samples. This follows what has been done in previous works (Penedo et al., 2024; Soldaini et al., 2024).

## 4 EXPERIMENTS

How good is the data produced by our pipeline? To assess this question we conduct experiments against the recently proposed FineWeb pipeline (Penedo et al., 2024). We do this be performing a data ablation experiment. Here, we train two language models on data produced by 1) our pipeline and 2) the FineWeb pipeline respectively. We then evaluate the language models as a proxy for evaluating the datasets and, in turn, the pipelines.

FineWeb uses trafilatura (Barbaresi, 2021) as HTML extractor and relies on quality filter sets from both C4 and Gopher, as well as some novel additions. A notable difference is the fact that trafilatura (in the setting used by FineWeb) produces plain text content, while SWEb format as Markdown. As mentioned in Section 3.2, we primarily retain formatting via Markdown *as a feature*, but note that this may also affect the learning behavior of the model. In this work however, we do not perform specific ablations to single out this particular factor. Please see Appendix F where we show side-by-side comparisons of trafilatura vs our extractor outputs.

How will Sweden be able to _____ itself in the international competition and strengthen its position as a leading knowledge nation? A first step is to look at the _____ that govern the allocation of state research funds.

**A** activate – knowledge
**B** mark – needs
Ⓒ assert – criteria
**D** entrust – institutions

Proper shoes are on the way out, while sneakers are spreading. The following _____ no longer causes any sensation: blazer, pleated trousers, and white sneakers.

**A** manner
**B** propriety
**C** ensemble
Ⓓ attire

Figure 5: Two examples from the HP-MEK task. Translated to English (originally in Swedish).

---

[5]We lowercased the text and removed non-alphabetic characters before creating shingles.

| Exp. Dataset | #Docs | #Tokens | Tokens/doc |
|---|---|---|---|
| SWEb | 32.3M | 25.2B | 779.7 |
| FineWeb | 19.2M | 15.8B | 820.3 |

Table 1: Stats of experimental datasets SWEb and FineWeb

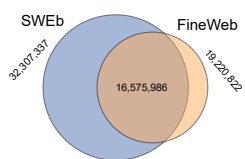

Figure 6: Venn diagram of documents in experimental SWEb and FineWeb datasets

## 4.1 BENCHMARK: HP-MEK

We investigated different benchmarks to evaluate the language models on. An appropriate benchmark should give good "early signals" of performance, in small model and data scales. For the Scandinavian benchmarks, the Scandeval suite (Nielsen, 2023) is commonly used. However, we found neither of its subtasks to be appropriate for this study, as the models didn't reach good enough performace.

Instead, we chose to develop an alternative benchmark based on the Swedish Scholastic Aptitude Test (Högskoleprovet), that we denote **HP-MEK**[6]. We download and extract the MEK (sentence completion) section of all available historical tests, and end up with a total of 460 examples. HP-MEK is a cloze style test, with masked portions of a provided passage. For each passage, four alternatives of the masked portions are available, see Figure 5. We evaluate a model by inserting each of the four alternatives into the passage, and pick the alternative with the highest joint log likelihood.

In our experiments, we see early and consistently increased performance as we train on successively more data, which speaks for it being a suitable indicator for performance at larger scales.

## 4.2 EXPERIMENTAL SETUP

We extract, filter and deduplicate the 2024-10 and 2024-18 Common Crawl snapshots using our pipeline to form an experimental dataset (SWEb). We also run the FineWeb pipeline on the same input documents (selected from Stage 1) to form a competing dataset (FineWeb). Table 1 compares the two and Figure 6 shows a Venn diagram of their document (url) sets.

We note that the SWEb pipeline extracts significantly more documents (+62%) and tokens (+60%) than FineWeb's pipeline. Most of FineWeb's documents are contained in SWEb, while relatively few are uniquely selected by FineWeb. Interestingly, FineWeb extracts slightly more tokens per document on average, despite SWEb containing additional Markdown formating tokens.

We split the two datasets in 90/10 train/test splits and tokenize using the GPT-SW3 tokenizer (Ekgren et al., 2024). Then, we train small language models on each training set respectively ($M_{SW}$ for SWEb and $M_{FW}$ for FineWeb), and use the Llama architecture with 1.82B parameters (including embeddings) with a 2048 sequence length, a global batch size of 2 million tokens and a cosine decay learning rate schedule. Each model is trained for 10,811 steps, which corresponds to one full epoch for SWEb, and 1.6 epochs for FineWeb. We checkpoint every 250 steps to evaluate progression throughout training.

## 4.3 RESULTS

In Figure 7, we show perplexity plots where each model is evaluated on the each of the two test sets. We can first note that $M_{SW}$ achieves lower perplexity on its own data than $M_{FW}$, i.e. SWEb seems "easier" to fit despite it being trained on more unique tokens. This could for example be due to the markdown formating, where markup tokens might be easier to predict. Secondly, $M_{SW}$ performs relatively better on FineWeb than $M_{FW}$ on SWEb. We speculate this could also be due to the markdown, where $M_{FW}$ gets more confused not having seen Markdown during training.

---

[6]Available at `https://huggingface.co/datasets/AI-Sweden-Models/HP-MEK`

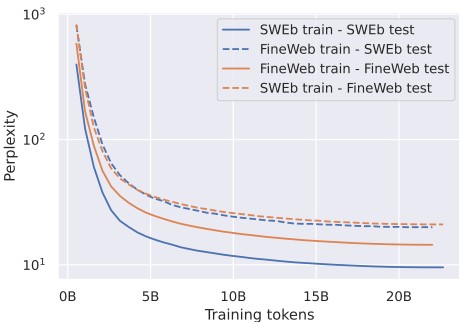

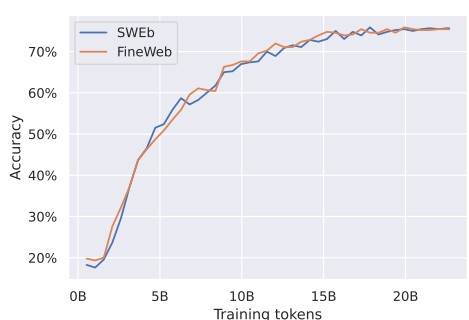

Figure 7: Perplexity cross-evaluation. The two models are evaluated on both SWEb and FineWeb test sets.

Figure 8: Learning curves. Performance of our two ablation models on HP-MEK throughout training.

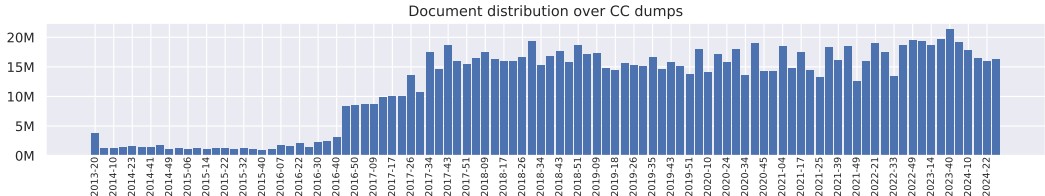

Figure 9: SWEb distribution over the Common Crawl snapshots.

Next, we evaluate $M_{SW}$ and $M_{FW}$ on HP-MEK, and plot learning curves in Figure 8. We can see that $M_{SW}$ closely matches $M_{FW}$ throughout the training, suggesting the two datasets are on-par with each other with regards to this task. This suggests that we are able to match the trafilatura extractor with just 1,380 annotated extraction samples, and at the same time reduce the complex filtering to only four simple quantities.

## 5 THE SWEB DATASET

We run our pipeline on 98 Common Crawl dumps, starting from 2013-20 until 2024-26, to produce the **Scandinavian Web (SWEb) dataset**. SWEb comprises a total of **1.01 trillion tokens**[7], distributed over **1.2 billion documents**, resulting in **3.6TB** of raw (UTF-8) text.

This makes SWEb the largest open Scandinavian dataset to date, an order of magnitude larger than the (to our knowledge) previously largest mC4 dataset. In Figure 9, we show the document distribution across the Common Crawl dumps. As we can see, the amount of Scandinavian content has been steady since around 2017, averaging about 50M documents per dump.

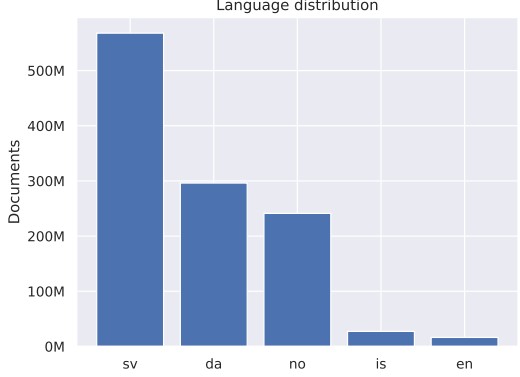

Figure 10: Language distribution over the SWEb dataset

To investigate the language distributon of SWEb, we use the fastText language indentification classifier by Joulin et al. (2016a;b). Among the four Scandinavian languages, Swedish is the dominating one with 48% of documents classified as Swedish, 26% as Danish and 20% as Norwegian, see Figure 10. Only 2.3% are classified as Ice-

[7]Using the GPT-SW3 (Ekgren et al., 2024) tokenizer

landic. A small portion of documents are classified as non-scandinavian after our content extraction, of which a majority is classified as English.

We release the SWEb dataset, the pipeline code, as well as our trained extractor model open source license, and hope this will further research and development of high performant Scandinavian LLMs. We also provide a datasheet detailing the dataset further in Appendix A.

## 6 DISCUSSION AND FUTURE WORK

Comparing to rule-based extractors such as trafilatura, our model based extractor offers greater flexibility as the desired extraction output is *demonstrated* instead of encoded as heuristics. Our work also highlights the data efficiency with which this can be done, i.e just 1,380 annotated examples in our case. However, this also comes with a cost. Running our model extractor for each document increases the compute required substantially over rule-based alternatives, which adds to these already compute-intensive workloads. In extracting SWEb, we consumed 20k AMD MI250X GPU-hours which is a significant amount, but comparing to the budgets required for training the downstream LLMs it is still negligable.

While training LLMs on larger datasets have shown to yield higher performance, a hypothesis is that there is only a subset of high quality documents that are behind the performance boosts. For example, in FineWeb-Edu, further filtering web data towards "educational content" is shown to significantly boosts performance in reasoning- and knowledge-intensive benchmarks. We see work on topic and content based filtering as a promising avenue for further refinement of SWEb towards particular LLM capabilities. This could potentially even be built into the extractor for more fine-grained control instead of as a binary post-hoc filter.

## 7 CONCLUSION

A major bottleneck for pre-training LLMs for smaller languages is the lack of large and high-quality open datasets. In this paper, we have presented the thus far largest open dataset for pre-training LLMs for the Scandinavian languages (Swedish, Danish, Norwegian and Icelandic). The dataset, which we call SWEb, comprises 1 trillion high-quality tokens in said four languages, and is openly shared in order to promote the development of LLMs for the Scandinavian languages. In creating SWEb, we have also developed a pipeline with a novel model-based text extractor that offers greater flexibility over the extraction process versus rule-based alternatives. We share both code and models for the novel text extractor openly. This paper has introduced a new benchmark for Swedish, which we use to compare models trained using our data with models trained using FineWeb, and we demonstrate that our data leads to models with performance on par with models trained from data using the state-of-the-art pipeline FineWeb.

## ETHICAL CONSIDERATIONS

**Handling Sensitive Content**   The SWEb dataset was created using publicly available Common Crawl data. During processing, efforts were made to filter out low-quality and irrelevant material, such as advertisements, spam, and repetitive text. While our approach focuses on a simplified and novel pipeline for text extraction, we do not implement specific filtering mechanisms for harmful or sensitive content. We acknowledge that pre-training data can influence the behavior of LLMs, potentially amplifying biases or generating harmful outputs. We encourage researchers and practitioners utilizing our pipeline and dataset to critically assess their data sources and apply appropriate filtering techniques based on their ethical and application-specific requirements. Future work should consider integrating robust content moderation strategies to mitigate risks associated with unfiltered pre-training data. Refer to the datasheet in the appendix for more details on the dataset's curation process.

**Privacy and Data Protection**   To address privacy concerns, some personally identifiable information (PII) such as email addresses and publicly visible IP addresses were removed using regex-based filters. While these methods are widely adopted in the field, we acknowledge their limitations and the

potential for residual PII. SWEb also adheres to Common Crawl's policies, which respect robots.txt and nofollow directives to avoid restricted data. For additional information on how PII and privacy were handled, refer to the datasheet in the appendix.

**Bias and Representation**    The SWEb dataset is derived from Common Crawl, which reflects the inherent biases of web-based data. These biases may arise from factors such as the overrepresentation of content from certain languages, regions, or demographics with greater internet access and technological literacy. Additionally, the dataset may include content from domains or sources that reflect specific viewpoints, and, as with any web-based resource, there is a risk of including disinformation or other misleading content. While the dataset aims to provide a valuable resource for Scandinavian languages, we encourage users to remain mindful of these potential biases and consider mitigation strategies during training and deployment. Further details on potential biases and dataset composition can be found in the datasheet in the appendix.

**Intended Use and Misuse Prevention**    The primary goal of SWEb is to support the research and development of Scandinavian language models. This dataset should not be used to train models that generate harmful content, misinformation, or other unethical outputs. Users are encouraged to implement safeguards and adhere to ethical AI development principles. Refer to the datasheet in the appendix for guidance on intended use and potential misuse.

**Takedown Policy**    We provide a mechanism for stakeholders to request the removal of specific content through our takedown policy. For more information on this policy, see the datasheet in the appendix.

ACKNOWLEDGMENTS

We gratefully acknowledge the Swedish Innovation Agency (Vinnova) for supporting this work. This work has also been supported by the Parallel computing center (PDC) at the Royal Institute of Technology in Stockholm, Sweden. We would like to express our deepest gratitude for providing the compute required for our experiments and building SWEb. We thank our annotators for their kind effort in helping out building our text extractor model.

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

## A  SWEB DATASHEET

We provide a datasheet inspired by Gebru et al. (2021):

| Motivation | |
|---|---|
| Purpose of the dataset | We want to encourage open research and development of LLMs in the Swedish, Danish, Norwegian and Icelandic languages. We build and release SWEb to promote this objective and to address the linguistic challenges specific to underrepresented Scandinavian languages, improving access to language technology in these regions. |
| Curated by | AI Sweden |
| Funded by | AI Sweden |
| **Composition** | |
| Data Fields | Each data instance contains: 1. The source URL 2. The original Common Crawl WARC file path 3. The WARC date 4. The extracted text content, in markdown format 5. The detected language (using fastText classifier) |
| Data Splits | We split SWEb based on Common Crawl dump, to allow for download based on time of crawl. We also include a default split containing the entire dataset. |
| Errors and noise | As outlined in this paper, we propose a novel model based approach to extract text from websites. However, the model is not perfect and non-relevant content as well as noise are sometimes also erroneously extracted. We try to filter such examples in our third pipeline stage, but despite our best effort such examples may sometimes slip through. |
| Offensive and toxic content | As we don't attempt to filter based on content or topic in this work, SWEb might contain content that can be percieved as offensive, threatening or otherwise toxic. When considering using this dataset, it is important to be aware of this and that further processing might be necessary depending on use case. |
| **Dataset Curation** | |
| Curation rationale | We use Common Crawl as it is the (to our knowledge) largest and most diverse open corpus available in the Scandinavian languages. |
| Source data | The Common Crawl source data consist of large amounts of webpages crawled from the open web. Common Crawl's crawlers has always respected `nofollow` and `robots.txt` policies. |
| Time frames of collected data | We use all Common Crawl scraped dating back to week 50 of 2013 and up to week 26 of 2024. |
| Data processing steps | See Section 3. |

| | |
|---|---|
| Annotations | Among the data fields, only the detected language can be considered "annotated" by us. |
| Personal & sensitive information | We anonymize email addresses and public IP addresses using regex patterns. |
| **Considerations for using the data** | |
| Social impact of dataset | With SWEb, our goal is to make LLM training more accessible to the machine learning community by: (a) making the dataset creation process more transparent, by sharing our entire processing setup including the codebase used (b) helping alleviate the costs of dataset curation, both in time and in compute, for model creators by publicly releasing our dataset with the community. |
| Bias and Representation | While the Common Crawl data gathers diverse text sources, biases present in the original content may still exist. Users should critically assess how these biases may affect model training and outcomes, especially in sensitive applications. It is recommended to implement bias-mitigation techniques during training and model development. |
| Model Misuse | When training models with this dataset, it is crucial to prevent harmful uses of the resulting models, such as generating misleading or dangerous content (e.g., disinformation, hate speech). Always consider the societal impact of deploying models trained on this data, and take precautions to implement appropriate safeguards. |
| **Distribution** | |
| Distribution platform | The dataset will be distributed on the Huggingface Hub |
| License | The data is released under the CC0 Creative Commons License. We make the following clarifications: 1. We do not warrant or guarantee any rights to the underlying data contained within this dataset. Users are solely responsible for validating and securing the appropriate rights and licenses for their specific intended uses. 2. This license applies only to the structure and compilation of the dataset as provided by us. We do not claim any database rights or ownership over the underlying data itself. Users must ensure compliance with any legal obligations, including those related to third-party content, copyrighted material, or personal information (PII) that may be contained in the underlying data. 3. With the release of this dataset, our goal is to promote and advance open research and the development of Scandinavian language models, showcase research outcomes as well as enable research validation. Open datasets are essential to fostering innovation and expanding knowledge in AI. We disclaim any responsibility for other uses, including commercial applications. Users are responsible for ensuring the legality of their usage, especially in cases involving copyrighted material. |

| Notice and take-down policy | Should you consider that our data contains material that is owned by you and should therefore not be reproduced here, please: |
|---|---|
| | 1. Clearly identify yourself, with detailed contact data such as an address, telephone number or email address at which you can be contacted. |
| | 2. Clearly identify the copyrighted work claimed to be infringed. |
| | 3. Clearly identify the material that is claimed to be infringing and information reasonably sufficient to allow us to locate the material. |
| | 4. You can reach us at datasets@ai.se |
| | We will comply to legitimate requests by removing the affected sources from the next release of the corpus. |

## B    MARKDOWN ANNOTATION DETAILS

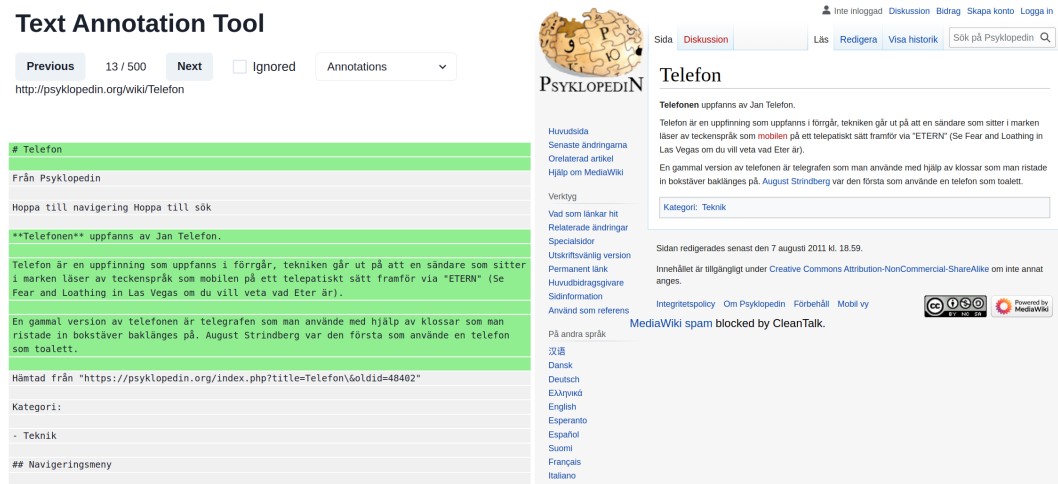

Figure 11: Our web based annotation tool. On the right side the original web page is displayed and on the left the corresponding markdown. Annotation is performed by selecting individual lines (marked green) that constitute the main content of the page.

We develop a web based tool that we use to annotate markdown documents, see Figure 11. The tool is used to annotate data for training and evaluating our text extractor model (Section 3.2.2).

The annotation was performed by the authors as well as additional lab colleagues, in total a group of 12 people. We started by jointly annotating a gold standard test set of 100 examples (web pages). This was useful to align and develop our annotation guidelines.

Next, we annotated a first set of 400 training examples and trained a first extractor model. This model served as a first baseline. We then iteratively annotated additional training data in batches of 300-500 examples, re-trained and re-evaluated after each iteration.

Judging what is "main content" in web pages is not always obvious however. When the evaluation didn't improve after a new batch of annotations, we developed a method for discovering "confusing" training examples in the new batch that we could jointly discuss and align on. For each example $x$ in the new training batch, we compute the loss $l^{M_n}(x, y) = \mathcal{L}(M_n(x), y)$, where $\mathcal{L}$ is the average over all BCE losses in the example and $M_n$ is the model trained on all batches including iteration $n$.

By comparing this loss to the corresponding loss under the *previous* model $M_{n-1}$, we get a measure of how "surprising" this example is:

$$\delta = l^{M_{n-1}}(x, y) - l^{M_n}(x, y) \tag{3}$$

Using this quantity, we could easily identify outliers and correct inconsistent annotations. By performing this post-annotation fix-up, we were able to improve performance on our test set, for each annotated batch of data.

## C CONTENT EXTRACTION ANNOTATION GUIDELINES

*The following description was provided to our annotators*

In the provided annotation tool, please select individual lines by clicking and dragging across the lines you want to select.

- Please look at the rendered web page on the right. We want to extract the **"main textual content"** of the current page.
- **No navigation** (menus, links, button texts etc) should be selected, except well formatted tables of content that link within the same page
- Include **headers** of main content
- If duplicate header, select the one closest to the main content
- Include **well formatted tables**
- Don't include content of **sidebars** that is unrelated to the main content
- It is OK to leave the whole document unselectede if there is no relevant content
- If there are many very **similar-looking pages**, they can be marked *Ignored* if they have already been annotated. Bad pages without any good content should **not** be ignored however.
- Include **comment sections** if there are any, but exclude navigation associated with those, e.g. *Svara / Rapportera inlägg* or similar.
- Keep **comment headings**
- If text is broken with e.g. "...", don't include
- Select **top heading** if it exists
- Keep **at most 2 consecutive newlines**
- Remove empty formatting lines (e.g **), except for dividers (———)
- Pages that are primarily "data" (e.g. tables, numbers) without much text should be unselected. There should be at least **three consecutive sentences** of text. This puts a somewhat high bar for product pages
- No HTML should be selected

# D    CONTENT EXTRACTOR ERROR ANALYSIS

To analyze the errors made by our context extractor, we use the line-level annotations of the 100 validation documents. For each line in each document, the line is predicted to be extracted or ignored. Therefore, for each document, we get a number of FP and FN line classifications. In Figure 12 we plot the distribution over number of FP and FN lines.

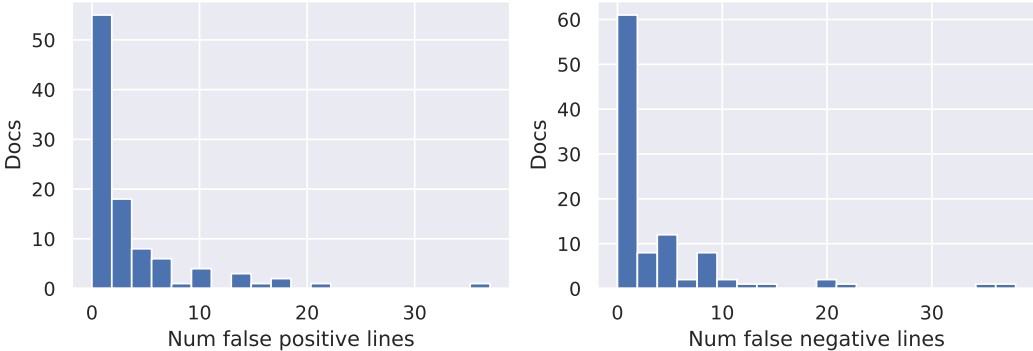

Figure 12: Distribution over number of False Positive/Negative lines, in the test set documents

We can see that the vast majority of documents have less than 10 wrongly classified lines. The document with the highest number of false negative lines is shown in Figure 13. In this case, the table of contents, some headings as well as the reference list were wrongly not extracted.

```
# Flyulykken i Kastrup 1947

**Flyulykken i Kastrup 1947** var en flyulykke, der fandt sted den 26. januar 1947 kl. 15:35 i Københavns Lufthavn, Kastrup, da en Douglas DC-3C (C-47) fra KLM Royal Dutch Airlines
styrtede ned umiddelbart efter starten efter en mellemlanding på vej fra Amsterdam til Stockholm. Alle 22 ombord omkom, blandt dem den svenske arveprins Gustaf Adolf, den nuværende kong
Carl XVI Gustafs far.

Flyulykken i Kastrup 1947

Vraget efter flyet

Ulykken skete som følge af, at flyet stallede under starten som følge af en glemt rorlås. Kaptajn på flyvningen var den erfarne 55-årige hollandske pilot Gerrit Johannis Geysendorffer.
Ulykken anses som den fjerdeværste danske flyulykke.

## Indholdsfortegnelse

- 1 Forløb
- 2 Omkomne
- 3 Noter
- 4 Se også
- 5 Eksterne links

## Forløb Rediger

Klokken 14.55 mellemlandede flyet i Kastrup for at medtage passagerer. På grund af kraftige vindstød blev der trods det korte ophold i Kastrup sat lås på flyets ror for at undgå, at disse
skulle blive beskadigede. Da flyet skulle lette igen, blev der formentlig ikke gennemgået tjekliste. Der blev heller ikke gennemført startcheck af flyets motorer. Disse to fejl medførte,
at piloterne ikke var opmærksomme på, at nogen havde glemt at fjerne låsen til højderoret.^(\[1\]) Resultatet blev, at da DC-3'eren lettede kl. 15.31, foretog flyet en brat stigning op
til ca. 50 meters højde, hvorefter flyet stallede og ramte jorden.^(\[2\]) Det havarerede fly brød straks i brand, og samtlige 22 ombordværende omkom.

- 

Flyets planlagte rute.

- 

En Douglas DC-3 fra KLM som det forulykkede fly.

## Omkomne Rediger

Ved starten fra Kastrup befandt sig ombord på flyet 6 besætningsmedlemmer og 16 passagerer; 11 mænd, tre kvinder og to børn. Af disse var syv danskere, fire svenskere, to franskmænd, en
amerikaner, en hollænder og en spanier.

Flyets kaptajn var den 55-årige Gerrit Johannes Geysendorffer, der var en erfaren pilot. Han var den første hollænder, der modtog et kommercielt flycertifikat og som siden 1921 havde
været tilknyttet KLM. Geysendorffer havde i 1927 som den første hollænder fløjet hele vejen til Batavia (nutidens Jakarta) i Hollandsk Ostindien og havde foretaget mere end 50 flyvninger
til den hollandske koloni, for hvilket han var blevet tildelt ordenen Orde van Oranje-Nassau.^(\[3\]) Geysendorffer blev certificeret til flyvning med DC-3 i 1937 og havde flere hundrede
flyvetimer med flyet.^(\[4\]) Han boede i Amsterdam med sin danske hustru Tofa Spandet, som han havde mødt, da han i en ung alder var nødlandet i Danmark.^(\[5\]) Flyets styrmand var Jan
Rietman (født 1918).^(\[6\])

Den svenske tronarving Gustaf Adolf var blandt syv passagerer på flyvningen fra Amsterdam til København. Gustav Adolf havde besøgt den hollandske Prins Bernhard i nogle dage, hvor de
havde været på vildsvinejagt sammen.^(\[7\]) Gustav Adolf rejste med sin adjudant, Albert Stenbock og var på vej til Stockholm.

Ved mellemlandingen i Kastrup steg tre passagerer af flyet og 12 nye passagerer steg om bord. Blandt passagererne, der steg ombord i Kastrup var den amerikanske operasanger og skuespiller
Grace Moore, der var på turne. Hun havde aftenen før optrådt i en fyldt KB Hal og var nu på vej til Stockholm for at give koncert. Med Grace Moore rejste diverse musikere m.v. og også den
danske skuespillerinde Gerda Neumann og dennes mand, direktør og filmproducent Jens Dennow.^(\[8\]) Flyets kendte passagerer havde pressens bevågenhed. Den svenske kronprins gav ikke
interviews til pressen, men lod sin adjudant besvare pressens spørgsmål, og Grace Moore og sine to medrejsende gæster, Gerda Neumann og Jens Dennow måtte besvare flere spørgsmål og lod
sig fotografere i lufthavnen.

- 

Gustaf Adolf

- 

Grace Moore

- 

Gerrit Geysendorffer
## Noter Rediger

1. ^
"Beskrivelse af ulykken på avition-saftet.net". Arkiveret fra originalen 8. juni 2015. Hentet 3. maj 2015.
2. ^ Alsing, Rolf; Lundh, Jonas, red. (1999). *1900-talet: en bok från Aftonbladet*. Stockholm: Aftonbladet med stöd av Statens skolverk. s. 144. ISBN 91-630-8939-4.`{{cite book}}`:
CS1-vedligeholdelse: Flere navne: editors list (link)Libris 7454380
3. ^ *Grace Moore and Heir to Throne Die in Crash.* I Los Angeles Times, 27. januar 1947, p. 2.
4. ^ Original-Luftfahrtuntersuchungsbericht, p. 3–4 Arkiveret 24. september 2015 hos Wayback Machine på hdekker.info (nederlandsk/hollandsk), hentet 3. maj 2015
5. ^ *Grace Moore dies in burning Plane* The New York Times, 27. januar 1947, p. 3.
6. ^ Original-Luftfahrtuntersuchungsbericht, p. 4–5 Arkiveret 24. september 2015 hos Wayback Machine, hdekker.info, (nederlandsk/hollandsk), hentet 3. maj 2015
7. ^ *Swedish Prince Killed In Air Wreck Crash After Taking Off At Copenhagen, Miss Grace Moore Among 22 Victims* The Times, 27. januar 1947, Nr. 50669, p. 4
8. ^ "Flykatastrofen i Kastrup 1947, \[\[Dines Bogø\]\], \[\[Amagerbladet\]\], 7. september 2009". Arkiveret fra originalen 24. december 2014. Hentet 3. maj 2015.
```

Figure 13: The extraction for https://da.m.wikipedia.org/wiki/Flyulykken_i_Kastrup_1947. Green lines are true positives and blue lines are false negatives. True negative lines are not shown for illustrative purposes.

# E    FILTERED EXAMPLES

We show examples of extracted documents that are *filtered out* by each of our four quality filters.

## E.1    CONTENT LENGTH < 100 CHARS

```
https://www.buskerudmynt.no/produkt/norske-mynter-etter-1874/norske-argangsmynter/50-ore/olav-v-1974-1991/
50-ore-1977-kv.-0
```

```
1  # 50 øre 1977 kv. 0
2
3  Tatt fra rull. Litt skjoldete mynt.
4
5  NOK5,00 inkl. mva.
```

```
https://www.ovedanielsson.se/2021/08/30/ohrmans-fick-inte-bygga-nytt-mot-torget/embed/
```

```
1  Öhrmans fick inte bygga nytt mot torget
```

```
https://jesper.nu/spel/achtung-die-kurve
```

```
1  # Achtung Die Kurve
```

## E.2    RATIO OF ALPHANUMERIC CHARACTERS < 0.4

```
https://www.innebandystats.se/statistik/219645/kevin-sandeback
```

```
1  |          | CL98IC                             | Juniorallsvenskan HJ18  |  14  |  16
       |   10   | **26**  |   0    |
```

```
https://nn.wikipedia.org/wiki/Kategori:Deltakarar_under_vinter-OL_1984_etter_Ãÿving
```

```
1  1896 ** Â·** 1900 ** Â·** 1904 ** Â·** 1906 ** Â·** 1908 ** Â·** 1912 ** Â·** ~~(1916)~~ ** Â·** 1920 ** Â·
       ** 1924 ** Â·** 1928 ** Â·** 1932 ** Â·** 1936 ** Â·** ~~(1940)~~ ** Â·** ~~(1944)~~ ** Â·** 1948 **
       Â·** 1952 ** Â·** 1956 ** Â·** 1960 ** Â·** 1964 ** Â·** 1968 ** Â·** 1972 ** Â·** 1976 ** Â·** 1980
       ** Â·** 1984 ** Â·** 1988 ** Â·** 1992 ** Â·** 1996 ** Â·** 2000 ** Â·** 2004 ** Â·** 2008 ** Â·**
       2012 ** Â·** 2016** Â·** 2020
2  **Vinter-OL**
3
4  Deltakarar etter **nasjon:**
5
6  1924 ** Â·** 1928 ** Â·** 1932 ** Â·** 1936 ** Â·** ~~(1940)~~ ** Â·** ~~(1944)~~ ** Â·** 1948 ** Â·** 1952
       ** Â·** 1956 ** Â·** 1960 ** Â·** 1964 ** Â·** 1968 ** Â·** 1972 ** Â·** 1976 ** Â·** 1980 ** Â·**
       1984 ** Â·** 1988 ** Â·** 1992 ** Â·** 1994 ** Â·** 1998 ** Â·** 2002 ** Â·** 2006 ** Â·** 2010 ** Â·
       ** 2014 ** Â·** 2018 ** Â·** 2022
7
8  Deltakarar etter **øving:**
```

```
https://historik.val.se/val/val2010/alkon/K/valdistrikt/12/80/0102/alderkon.html
```

```
1  | ----------------------- | ----: | ----: | ----: | ----: | ----: | ----: | ----: | ----: | --: |
       ----------------: | ----------------: | ----: | --: | ------: | ------: | ------------------: |
       ------------------: | --------------------: | --------------------: |
2  | Gamla staden, Stortorget |  1533 | 24,0% |   380 | 43,0% |   659 | 20,4% |   312 | 11,9% | 182 |
                 4,4% |              68 | 52,6% | 806 |   47,4% |   727 |                13,8% |
             212 |                            |                 |
3  | Summa                 |  1533 | 24,0% |   380 | 43,0% |   659 | 20,4% |   312 | 11,9% | 182 |
                 4,4% |              68 | 52,6% | 806 |   47,4% |   727 |                13,8% |
             212 |                            |                 |
4
5  http://www.val.se
```

## E.3    HEADINGS PER NON-HEADING WORD > 0.05

```
https://www.sahlgrenska.se/for-dig-som-ar/vardgivare/laboratoriemedicin/analyslistan/
specialprover-cytologisk-diagnostik/16648.html/
```

```
1  # Glaskropp
2
3  # Glaskropp cytologisk diagnos
4
5  ### Synonymer
6
```

```
 7  Specialprover, cytologisk diagnostik
 8
 9  ## Provtagningsanvisning
10
11  ### Provmaterial
12
13  ### Rör el. motsv
14
15  10 ml rör med gul kork, konisk botten, steril (för mindre mängder material) eller Burk ca 40 ml m tä
        tslutande lock, sterilt
16
17  ### Provtagning
18
19  Enligt inremitterande kliniks regler Provet skall snarast efter sköljningen transporteras till Cytologen.
20  Ofixerade vätskor ska föranmälas och lämnas direkt till lab. personal före kl 14.00.
21
22  ### Transport
23
24  Transport ska ske omgående till Laboratoriet för klinisk patologi och där lämnas direkt till provinlä
        mningen.
```

https://folk.sunnhordland.no/publications/211951

```
 1  # Olive Buchvold Juvik
 2
 3  Gullet vårt Olive «snat 2 år» «Ja, vennen, på lørdag 18. nov, fyller du 2 år» Me gratulerer så masse\!
        Kjempe gla' i deg. Klem fra tanter, onkler, besteforeldre og oldeforeldre.
 4
 5
 6
 7  ## Go'ungen (0-12 år)
 8
 9
10
11  ### Nora Silden Fredheim
```

https://start.arcada.fi/sv/kurser/303000/2021-2022/IA-2-004/0

```
 1  ## Kursens undervisningsperiod
 2
 3  3 (2022-01-01 till 2022-03-13)
 4
 5  ## Nivå/kategori
 6  ## Cykel/nivå
 7  Yrkeshögskoleexamen
 8
 9  ## Rekommenderat studieår
10
11  1
12  ## Omfattning
13
14  5 sp
15
16  ## Kompetensmål
17
18  I denna studieenhet står följande kompetenser i
19  fokus:
20  \- Kompetens inom serverprogrammering
21  \- Kompetens inom databashantering och lagring av
22  data
23  \- Kompetensen att skapa dynamiska applikationer
24
25  ## Läranderesultat
26
27  Efter avlagd studieenhet:
28  \- Du behärskar programmering med
29  PHP (Kunskap)
30  \- Du ser skillnaden mellan statiska, interaktiva
31  och dynamiska webbsidor (Kunskap)
32  \- Du kan hantera filer från klienten och på
33  servern (Kunskap)
34  \- Du kan bygga dynamiska webbappar (Färdighet)
35  \- Du kan lagra data säkert i en databas
36  (Färdighet)
37  \- Du inser problematik och lösningar kring att
38  lagra känslig information om en användare
39  (Förhållningssätt)
40  \- Du uppfattar olika sätt att överföra och lagra
41  data och dess koppling till säkerhet och
42  prestanda (Förhållningssätt)
43  \- Du uppfattar din makt och ditt ansvar som
```

## E.4   Unigram entropy < 3.0

https://hastkatalogen.se/content/horse/info.php?id=31999

```
1  # Catinkaox
2
3  ## Arabiskt Fullblod
4
5  Catinka är ett sto som föddes 1976 i Sverige.
6
7
8
9    - Ras: Arabiskt Fullblod
10   - Kön: Sto
11   - Färg: brun
12   - Stofamilj:
13     |                                              |
```

https://www.nilssonsilammhult.se/hallmobler/ida-skohorn-ek/

```
1  # Ida skohorn ek
2
3  430 kr
4
5  Ida skohorn i oljad ek från småländska Westroth. Tillverkad i formpressat trä. En fin detalj till hallen\!
6
7
8
9  Ida skohorn ek mängd
10
11 # Ida skohorn ek
12
13
14
15 Ida skohorn i oljad ek från småländska Westroth. Tillverkad i formpressat trä. En fin detalj till hallen\!
16
17 430 kr
```

https://kaldarsel.is/author/heidbjort-arney/

```
1    -
2
3  ## Leikjanámskeið 10. júlí
4
5  Höfundur: Heiðbjört Arney|2017-07-12T10:01:09+00:0012. júlí 2017|
6
7    -
8
9  ## Veisludagur runninn upp
10
11   -
12
13 ## Dvalarflokkur
14
15 Höfundur: Heiðbjört Arney|2017-06-28T10:15:51+00:0028. júní 2017|
16
17   -
18
19 ## Leikjanámskeið 2
20
21 Höfundur: Heiðbjört Arney|2017-06-21T13:07:08+00:0021. júní 2017|
```

# F    COMPARING OUR MODEL EXTRACTOR VS TRAFILATURA

We compare our model based extractor vs trafilatura (in the settings used by FineWeb).

```
https://www.ark.no/produkt/boker/dokumentar-og-faktaboker/eksil-9788202253912
```

## Trafilatura

```
1  Innbundet
2  2005
3  Norsk, Bokmål
4  «Denne boken dreier seg om eksil og dannelse.
5  Lesning av Dante ga meg en italiensk regel:
       Dannelse oppstår alltid og bare i eksil.
       Det vesle som fins av dannelse i Norge,
       dannes ut fra evnen til distanse i et
       livsnødvendig indre eller ytre eksil.
       Dannelse er det motsatte av turisme. Slik
       førte min selvomsorg meg stadig mer inn
       og ut av norsk kultur (og underholdning)
       til jeg ble uhelbredelig gudløs og partil
       øs i en vag, men livslang interesse for
       eksemplariske flyktninger og forrædere
       fra Klosterlasse til Asbjørn Sunde.»
6  (fra Georg Johannesens forord)
7  Klikk&Hent
8  På lager i 8 butikker
9  Nettlager Sendes normalt innen 1-2 virkedager
10 Bytt i alle våre butikker
11 -
12 Klikk og hent
13 -
```

## Model extractor (ours)

```
1  # Eksil - om klosterlasse og andre eksempler
2
3  Av Georg Johannesen
4
5  «Denne boken dreier seg om eksil og dannelse.
       Lesning av Dante ga meg en italiensk
       regel: Dannelse oppstår alltid og bare i
       eksil. Det vesle som fins av dannelse i
       Norge, dannes ut fra evnen til distanse i
       et livsnødvendig indre eller ytre eksil.
       Dannelse er det motsatte av turisme.
       Slik førte min selvomsorg meg stadig mer
       inn og ut av norsk kultur (og
       underholdning) til jeg ble uhelbredelig
       gudløs og partiløs i en vag, men livslang
       interesse for eksemplariske flyktninger
       og forrædere fra Klosterlasse til Asbjørn
       Sunde.» (fra Georg Johannesens forord)
```

```
https://gipomusic.se/track/vad-dom-an-tror/
```

## Trafilatura

```
1  Hur dom än
2  Färgerna är blekare än igår
3  tiden är för mörk för att vi ska kunna le
4  jag vill inte höra deras röst mer
5  Illusioner av tröst som drar mig ner
6  Hur de än sargar oss så ska vi hålla hand
7  Halva jävla världen är i brand
8  O hur dom än sänker oss så ska vi skrika högst
9  ett nej är alltid ett nej
10 Vart vi än går ser vi ner
11 aldrig mer igen, aldrig mer
12 hela tiden får vi säga till
13 ljusen runtomkring står bara still
14 Hur de än sargar oss så ska vi hålla hand
15 Halva jävla världen är i brand
16 O hur dom än sänker oss så ska vi skrika högst
17 ett nej är alltid ett nej
18 En vacker stråle som försvann
19 innan det blev mörkt
20 innan det blev kallt
21 Och om det var dina skrik som inte hördes
22 eller var din dotter som fördes iväg
23 hur skulle det kännas, hur skulle däää
24 Hur de än sargar oss så ska vi hålla hand
25 Halva jävla världen är i brand
26 O hur dom än sänker oss så ska vi skrika högst
27 ett nej är alltid ett nej
```

## Model extractor (ours)

```
1  # Vad dom än tror
2
3  Text: Clara Rudelius
4
5  https://gipomusic.se/wp-content/uploads
       /2014/10/04\_Vad-dom-än-tror.mp3
6
7  **Hur dom än**
8
9  Färgerna är blekare än igår
10 tiden är för mörk för att vi ska kunna le
11 jag vill inte höra deras röst mer
12 Illusioner av tröst som drar mig ner
13
14 Hur de än sargar oss så ska vi hålla hand
15 Halva jävla världen är i brand
16 O hur dom än sänker oss så ska vi skrika högst
17 ett nej är alltid ett nej
18
19 Vart vi än går ser vi ner
20 aldrig mer igen, aldrig mer
21 hela tiden får vi säga till
22 ljusen runtomkring står bara still
23
24 Hur de än sargar oss så ska vi hålla hand
25 Halva jävla världen är i brand
26 O hur dom än sänker oss så ska vi skrika högst
27 ett nej är alltid ett nej
28
29 En vacker stråle som försvann
30 innan det blev mörkt
31 innan det blev kallt
32
33 Och om det var dina skrik som inte hördes
34 eller var din dotter som fördes iväg
35 hur skulle det kännas, hur skulle däää
36
37 Hur de än sargar oss så ska vi hålla hand
38 Halva jävla världen är i brand
39 O hur dom än sänker oss så ska vi skrika högst
40 ett nej är alltid ett nej
41
42 ## Albumspår
```

https://fjordsaga.com/no/turer/2-i-kjolvannet-av-bilfergen-vaage-norvik

## Trafilatura

```
1  Turinformasjon
2  Tur fra Vågstranda til Norvika i Eidsbygda
       Rauma i kjølvannet av bilfergen Vaage-
       Norvik som gikk der fra 1930 til 1945.
3  Vei til Åndalsnes ble til stor del bygget
       ferdig av okkupasjonsmakten under andre
       verdenskrig og veien åpnet rundt
       tidspunktet for freden i 1945.
4  Denne fergen ble bygget av samme båtbygger som
        båten vi går turene med og det blir
       fortalt historie rundt dette samt
       hendelsene rundt den tragiske ulykken i
       oktober 1942 hvor Kultur og
       Propagandaminister i Quisling regjeringen
        Gulbrand Lunde m/frue omkom ved
       fergekaien på Vaage.
5  Turprisen er oppgitt pr passasjer basert på
       max antall. Ta kontakt for alternativer
       og evt allergier.
6  Eventuelt servering ombord!
7  1. Rik tomat/chili basert kremet fiskesuppe
       servert m/nybakt brød, dessert (Tilslørte
        bondepiker) og kokekaffe. Kr. 350.-
8  Lunsjpakke fra Braud Håndverksbakeri Vestnes:
9  2. Påsmurt bagett med ost & skinke +
       kanelbolle alt. solskinnsbolle. Kr. 110.-
10 3. Påsmurt bagett med kylling & karri +
       kanelbolle alt. solskinnsbolle. Kr. 120.-
11 4. Pastasalat med kylling og karri. Kr. 175.-
12 Mineralvann og annen drikke fås kjøpt separat
       om bord.
13 5 Timer
14 -
15 Maks. Passasjerer: 12
16 -
17 Vestnes
18 -
```

## Model extractor (ours)

```
1  # I kjølvannet av Bilfergen Vaage-Norvik
2
3  ### 1 100 NOK pr passasjer
4
5  ## Turinformasjon
6
7  Tur fra Vågstranda til Norvika i Eidsbygda
       Rauma i kjølvannet av bilfergen Vaage-
       Norvik som gikk der fra 1930 til 1945.
8
9  Vei til Åndalsnes ble til stor del bygget
       ferdig av okkupasjonsmakten under andre
       verdenskrig og veien åpnet rundt
       tidspunktet for freden i 1945. Denne
       fergen ble bygget av samme båtbygger som
       båten vi går turene med og det blir
       fortalt historie rundt dette samt
       hendelsene rundt den tragiske ulykken i
       oktober 1942 hvor Kultur og
       Propagandaminister i Quisling regjeringen
        Gulbrand Lunde m/frue omkom ved
       fergekaien på Vaage.
10
11 Turprisen er oppgitt pr passasjer basert på
       max antall. Ta kontakt for alternativer
       og evt allergier.
12
13 **Eventuelt servering ombord\!**
14
15 1\. Rik tomat/chili basert kremet fiskesuppe
       servert m/nybakt brød, dessert (Tilslørte
        bondepiker) og kokekaffe. Kr. 350.-
16
17 Lunsjpakke fra Braud Håndverksbakeri Vestnes:
18 2\. Påsmurt bagett med ost & skinke +
       kanelbolle alt. solskinnsbolle. Kr. 110.-
19 3\. Påsmurt bagett med kylling & karri +
       kanelbolle alt. solskinnsbolle. Kr. 120.-
20 4\. Pastasalat med kylling og karri. Kr. 175.-
21
22 Mineralvann og annen drikke fås kjøpt separat
       om bord.
23
24   - **5 Timer
25   - **Maks. Passasjerer: 12
26   - Avgang:Vestnes
27   - Turspråk:Engelsk, Norsk
```

