# OpenReview forum: "SWEb: A Large Web Dataset for the Scandinavian Languages"
_ICLR.cc/2025/Conference — ICLR 2025 Poster_

### Official Review · Reviewer_4Vny · 2024-11-01

**Soundness:** 3
**Presentation:** 3
**Contribution:** 2
**Rating:** 5
**Confidence:** 3

**Summary:**

The paper presents a new pretraining corpus for Scandinavian languages:Swedish, Danish, Norwegian, and Icelandic. To create this data, they propose a new pipeline that uses a model-based text extractor trained with a small amount of human-labelled data on what is considered the main content in a markdown version of an HTML webpage. They contrast their proposed approach against the FineWeb pipeline, which extracts plain text directly from HTML pages. They further introduce a new cloze style test with a dataset, HP-MEK consisting of 460 examples based on the Swedish Scholastic Aptitude Test (Högskoleprovet),  to benchmark pre-trained performance and show that on a small dataset+model setting, their extractor can match the performance of FineWeb filter, despite being simpler in complexity and using an extractor that is only trained with 1380 human-labeled examples.

Once validated, they use this pipeline to create SWEb, which includes 1.01 trillion tokens.

**Strengths:**

1. The authors present a novel model-based extractor, trained with 1380 human-labelled examples, to identify the main context from documents (HTML converted to markdown).
2. They introduce a new task and benchmark to validate their pipeline against a baseline, FineWeb and show that their pipeline, while being simpler, results in close performance on this task.
3. They present a new pretraining corpus of 1.01 trillion tokens for Scandinavian languages, which will be a valuable resource to the community.

**Weaknesses:**

1. It is unclear why the authors chose to retain the markdown tags for pretraining (section 4). They also do not explain much about why HP-MEK is a reasonable benchmark for the pretraining setting.
2. No direct analysis is presented on the efficacy of the model-based extractor, given that it is one of the main component of the paper apart from reporting an F1 score. The downstream application in section 4 is useful but it doesn't say much about what this extractor does or is capable of filtering.
3. The dataset pipeline only includes filters like content length, # of alphanumeric characters, and unigram entropy for quality filtering. However, it would have been useful if there was any direct consideration of what is considered high-quality data in the context of pretraining and if additional checks were made in place about safety and fairness of representation in the dataset.

**Questions:**

See weakenesses.

**Details Of Ethics Concerns:**

The paper presents a new pretraining dataset for SWEb which I believe is a very useful resource for the community but there is no discussion of ethical aspects of this dataset and if and whether measures where considered to prevent toxic and harmful content in the curation process.

---

> ### Author Response · Authors · 2024-11-20
>
> Thank you for the valuable feedback on our manuscript. We appreciate the opportunity to clarify and enhance our work based on your insights. Below, we provide our detailed responses to each of your comments:
>
> 1. Retention of Markdown Tags (Section 4): We chose to retain markdown tags in the pretraining data as these tags serve as structural markers that can help the model learn nuanced text segmentation and formatting cues. Markdown tags can signal important elements like headers, lists, and emphasized text. We recognize the implications this has on LM performance is not explored in our work, but also want to point out that the markdown formatting can easily be removed should a dataset user want it. We will update Section 4 to make this more explicit.
>
> 2. Analysis of the Model-Based Extractor: We have tried to evaluate the extractor efficacy both intrinsically (e.g. the F1 score you point out) and extrinsically (e.g. training LMs and evaluate/compare them). We also include a few qualitative samples for comparison in Appendix E. We acknowledge however that the intrinsic analysis could be further strengthened by for example an error analysis.
>
> 3. Consideration of High-Quality Data, Safety, and Fairness: We agree that ensuring high-quality data in the context of pretraining is crucial. In FineWeb-EDU, it is for example suggested that "educational"-like content can be particularly beneficial in LM pre-training. In this work, we chose to focus on quantity and recall and allow for subsequent work to further filter and optimize the quality of the dataset based on content and topic. Regarding bias and fairness, we partially address this in the datasheet (Appendix A).
>
> We hope these clarifications and proposed revisions meet your expectations. Thank you for your thoughtful feedback and for helping us improve our manuscript.

---

> > ### Comment · Reviewer_4Vny · 2024-11-27
> > **Response to rebuttal**
> >
> > 1. Thanks for the response, without proper ablation, its hard to say whether the markdown structure truly helps the LLM learn better.  Yes, please update section 4.
> > 2. Do you plan to include any additional error analysis in the final version? if yes, could you mention what that would be?
> > 3. Thanks for pointing out the datasheet.

---

> > > ### Author Response · Authors · 2024-11-27
> > >
> > > Thank you for your response.
> > >
> > > > Do you plan to include any additional error analysis in the final version? if yes, could you mention what that would be?
> > >
> > > Yes, we are aiming to add an additional error analysis appendix in the final version with both stats and examples from the test set where the extraction has failed.

---

### Official Review · Reviewer_dCGi · 2024-11-03

**Soundness:** 4
**Presentation:** 3
**Contribution:** 3
**Rating:** 6
**Confidence:** 3

**Summary:**

This work introduces a new pretraining dataset for Scandinavian languages, as well as a cloze-style evaluation dataset for Swedish language models.

**Strengths:**

- Usefulness and relevance: Improving the quality and assessing that this quality has indeed improved for LLMs targeting languages other than English is a clearly relevant topic that will have uses even outside of academia.

- Open-source: Authors promise the release of the dataset, benchmark, and utils used to generate/evaluate them.

- Technical details and examples: The paper features many detials about the implementation. Even if it was not open-source, I feel fairly confident that this work could be majorly reproduced.

**Weaknesses:**

While it's a sound contribution, I'm not sure ICLR is the right venue for this work. It lacks algorithmic or theoretical novelty, and it's rather a (very good) application of well-known NLP principles to process a new dataset for specific languages.

For example, the heuristics mentioned in the paper are very similar to "old" related work, e.g. [1]

https://arxiv.org/abs/1912.07076

**Questions:**

The pretraining dataset supports Scandinavian languages, but the benchmark targets Swedish. With the Swedish benchmark as a starting point, how much work would it be to generate similar benchmarks for other Scandinavian languages?

---

> ### Author Response · Authors · 2024-11-20
>
> Thank you for reviewing our work and for providing constructive feedback.
>
> While our work does not focus on theoretical developments, we argue our model based text extraction is a novel method in constructing pre-training datasets for language modeling. As we also release SWEb along with the HP-MEK benchmark, we believe it matches the "datasets and benchmarks" topic, listed under Subject Areas in ICLR's call for papers.
>
> As you point out, the evaluation (i.e. HP-MEK) is only in Swedish and not the other Scandinavian languages. As other benchmarks for these languages (e.g. Scandeval) didn't provide good signal at this small scale, we prioritized constructing an appropriate benchmark in the majority language (Swedish). Following up with corresponding tests in Danish, Norwegian and Icelandic is a top priority in subsequent work. A challenge however is that corresponding SAT tests in Norway, Denmark and Iceland does not exist, so such benchmarks requires a different construction method.

---

> > ### Comment · Reviewer_dCGi · 2024-11-26
> > **Answer to authors**
> >
> > Thanks for your answer. This makes sense. I'll update my rating.

---

### Official Review · Reviewer_SsWA · 2024-11-03

**Soundness:** 2
**Presentation:** 2
**Contribution:** 3
**Rating:** 6
**Confidence:** 4

**Summary:**

The paper describes the creation of a currently largest Scandinavian data set for training language models.
A new method for collecting texts from the web is proposed, based on an encoder model, and compared with a rule-based method.

**Strengths:**

The data set will be a valuable tool for researching language models and understanding their properties for Scandinavian languages, which are all under-resourced and under-investigated.

**Weaknesses:**

Some important points about the evaluation are not fully clear.

"Benchmark HP-MEK" is unclear: what was the motivation to created this test set and what was exactly evaluated exactly on it?
The new text extraction method or language models trained on the extracted texts?
For language models (section 4.2) it is said that there is 90/10 training/test splitting.
Therefore it is not clear what was involved in evaluations (text extractor or language models or both) and how (on which test set/s).

**Questions:**

182: What were are the annotators exactly instructed to do? How did they mark the main content?

195 isn't => is not

200-202: what are the "binary line annotations" exactly?
Can they be seen in the example in Listing 1?

300-301: What is "trafilatura"? Explanation citation?

Why is markdown better than plain text?

308: alternative benchmarks => alternative to what?

309: to evaluate performance on => performance of what? Of the proposed text extraction method?


315: didn't => did not

351: which model?

376: on the two test sets -- which ones? the one from 90/10 split and the other is HP-MEK?


Figure 1: what is "en"? It is not discussed in the text.



426: as the desired extraction output is demonstrated instead of encoded as heuristic => the meaning of the sentence is unclear; what does "demonstrated" means? What does "encoded as heuristic" means?

---

> ### Author Response · Authors · 2024-11-20
>
> We sincerely thank you for providing constructive feedback on our work, and wish to help clarify your questions.
>
> To compare our data extraction method versus the FineWeb method, we opt to train a small LM on a data sample extracted using the two methods (Table 1) and evaluate/compare them. As the available benchmarks for our targeted languages (e.g. Scandeval) didn't provide good signal (=close to random accuracy for both FineWeb and our method) at this small scale, we chose to develop a benchmark on which models progressed smoothly also at this small scale.
>
> This means we trained *two* LMs, one for each of the experimental datasets in Table 1. In Figure 7, we compare both models' perplexity on the experimental datasets' test sets (that is, four evals). In Figure 8, the two models are evaluated on HP-MEK.
>
> We hope this is clarifying. If not, please let us know!
>
> Regarding your questions:
>
> > 182: What were are the annotators exactly instructed to do? How did they mark the main content?
>
> Please find the exact annotator guidelines in Appendix C. The annotators marked the main content by selecting lines in the annotation tool, see Figure 11.
>
> > 200-202: what are the "binary line annotations" exactly? Can they be seen in the example in Listing 1?
>
> The binary line annotations refer to each line being either marked as constituting "main content" (green) or not (gray) in the annotation tool, see Figure 11. We will add a reference in the next paper revision, thanks!
>
> > 300-301: What is "trafilatura"? Explanation citation?
>
> We have a reference (https://aclanthology.org/2021.acl-demo.15/) in the Related Work section, but will add it here too!
>
> > Why is markdown better than plain text?
>
> We chose to retain formatting as a feature, to allow downstream LMs to explicitly model structural elements such as headings, text style (bold/italic) and tables. We recognize the implications this has on LM performance is not explored in our work, but also want to point out that the markdown formatting can easily be removed should a dataset user want it.
>
> > 308: alternative benchmarks => alternative to what?
>
> Good question. We specifically refer to the Scandeval suite (https://aclanthology.org/2023.nodalida-1.20/) and will add a reference!
>
> > 309: to evaluate performance on => performance of what? Of the proposed text extraction method?
>
> The performance of downstream LMs trained on data produced by our method. We agree this is not clear and will clarify this in the next revision, thanks!
>
> > 351: which model?
>
> We agree also this comes out unclear. We refer to the two LMs described on 367, and will make sure to clarify this in the next revision.
>
> > 376: on the two test sets -- which ones? the one from 90/10 split and the other is HP-MEK?
>
> We split both of the two experimental datasets in Table 1 in 90/10, and here we refer to those test sets.
>
> > Figure 1: what is "en"? It is not discussed in the text.
>
> Good point. "en" refers to English, as a small portion of documents are classified as non-scandinavian after our content extraction. We will add a note about this in the text!
>
> > 426: as the desired extraction output is demonstrated instead of encoded as heuristic => the meaning of the sentence is unclear; what does "demonstrated" means? What does "encoded as heuristic" means?
>
> What we mean is that with model based extraction, we "demonstrate" explicitly in the annotated training data what we consider "main content" and not. This is in contrast to defining rules and heuristics of what "main content" is, as is the case with e.g. Trafilatura.
>
>
> We hope these clarifications find you well. Please don't hesitate if any questions or remarks remain!

---

> > ### Comment · Reviewer_SsWA · 2024-11-26
> > **Response to the authors**
> >
> > Thank you for all clarifications!
> > If all these revisions are applied in the submission, it will be acceptable for publication, therefore I will raise my recommendation score.

---

> > > ### Author Response · Authors · 2024-11-27
> > >
> > > Thank you! The revisions should now be applied in the latest submission.

---

### Official Review · Reviewer_picV · 2024-11-03

**Soundness:** 3
**Presentation:** 3
**Contribution:** 3
**Rating:** 8
**Confidence:** 5

**Summary:**

The paper introduces SWEb, the largest pretraining dataset for Scandinavian languages, containing over one trillion tokens across Swedish, Danish, Norwegian, and Icelandic. SWEb aims to address the scarcity of large-scale, high-quality datasets specifically tailored for these languages. To create this dataset, the authors develop a model-based text extraction pipeline that enhances efficiency and reduces complexity compared to rule-based methods. Key contributions include:

- SWEb Dataset: An extensive web dataset of Scandinavian languages with over one trillion tokens, surpassing existing resources by an order of magnitude.
- Model-Based Extraction Pipeline: A novel, data-driven text extraction model that effectively filters high-quality content, yielding about 60% more usable tokens than previous approaches like FineWeb.
- Swedish Benchmark (HP-MEK): A new cloze-style benchmark for Swedish, derived from the Swedish Scholastic Aptitude Test, to evaluate language models trained on SWEb and demonstrate competitive performance against models trained on FineWeb.

The authors openly release the SWEb dataset, extraction pipeline, and the HP-MEK benchmark to support further research and development in Scandinavian language modeling​

**Strengths:**

Originality

The SWEb dataset is original in its approach to handling Scandinavian languages. The authors create a model-based extraction process that moves away from rule-heavy, manual extraction methods, simplifying the pipeline. They also introduce HP-MEK, a benchmark specific to Swedish, which adds value by providing a relevant evaluation tool for Scandinavian models.

Quality

The quality of the work is evident in the detailed steps of the SWEb pipeline. The authors carefully build a process to select and clean high-quality data, resulting in 60% more usable tokens than previous approaches like FineWeb. They validate the dataset with clear metrics, comparing models trained on SWEb and FineWeb to show the effectiveness of their extraction model.

Clarity

The paper is organized well, making each pipeline stage easy to understand. Diagrams and examples help clarify complex steps like content extraction and filtering. The authors document their choices for filtering and quality control, making the approach easier to follow and replicate.

Significance

SWEb is significant because it makes a high-quality, large-scale dataset available for Scandinavian languages, which traditionally have fewer resources. This dataset and the HP-MEK benchmark can help researchers build better models for these languages, making SWEb a useful resource for Scandinavian language research.

**Weaknesses:**

1. Limited Applicability Beyond Scandinavian Languages
Weakness: The SWEb pipeline is tailored specifically for Scandinavian languages, potentially limiting its scalability or adaptability to non-Scandinavian or low-resource languages. This narrow focus may reduce the general utility of SWEb’s approach in multilingual or global settings where language resources are scarcer.

Recommendation: It would be beneficial to discuss adapting the pipeline to other language families or the performance challenges in non-Scandinavian languages. Providing generalization strategies, such as multilingual training or enhanced language detection techniques, could expand SWEb’s relevance. Including preliminary results or small-scale tests on other language groups would further strengthen the paper’s broader applicability.

2. Reliance on Manual Annotation in Model Training
Weakness: Although SWEb’s model-based extractor is innovative, it relies on manually annotated data (1,380 samples) for training the extraction model, which may be resource-intensive and impractical for other languages or domains. Annotating thousands of samples for every new language could become a bottleneck, especially in low-resource contexts.

Recommendation: Introducing semi-supervised or weakly supervised learning approaches to reduce the dependency on manually annotated data could improve scalability. Alternatively, SWEb could consider leveraging existing rule-based systems or transfer learning from similar languages to jumpstart the model training in new language settings, potentially improving data efficiency.

3. Evaluation Restricted to Swedish Benchmark
Weakness: The evaluation uses only a Swedish benchmark (HP-MEK) to compare SWEb against FineWeb. While effective for a Swedish-specific evaluation, this choice does not cover other Scandinavian languages in the dataset (e.g., Danish, Norwegian, Icelandic), making it difficult to generalize the effectiveness of SWEb’s extraction pipeline across the entire language set.

Recommendation: Expanding the evaluation to include benchmarks for other Scandinavian languages or adapting HP-MEK for Danish, Norwegian, and Icelandic could enhance the assessment’s robustness. Providing a language-specific performance analysis would offer insights into whether the extraction model’s quality varies across languages and help optimize future language-specific models.

4. Lack of Qualitative Analysis of Extracted Content
Weakness: The quantitative metrics (e.g., token count, perplexity, accuracy) demonstrate SWEb’s improvements but lack a qualitative assessment of the extracted text's relevance or coherence. Without this, it’s challenging to understand how well the extraction model preserves the content’s intended meaning or cultural context, especially when removing ads and navigation elements.

Recommendation: Including a qualitative evaluation of content extracted by SWEb compared to FineWeb, such as reader surveys or manual inspection of content fidelity, would offer a deeper understanding of its cultural and contextual accuracy. This analysis would help validate the extractor’s ability to maintain high content relevance, especially for Scandinavian-specific terms, phrases, or topics that might be lost during processing.

5. Limited Error Analysis in Content Extraction
Weakness: The paper does not provide an in-depth error analysis for the types of errors encountered during extraction, such as failures to remove advertisements, incorrect content classification, or issues in handling specific webpage structures. This gap makes it difficult to assess the limitations of the extraction model and how it could be improved.

Recommendation: An error analysis that categorizes extraction mistakes (e.g., missed ads, incorrectly retained menu items, or misclassified headers) would clarify the model’s boundaries and suggest refinements. Detailing any challenges in handling regional dialects or slang in Scandinavian languages would identify areas for improvement in future iterations of SWEb.

6. Computational Expense of the Extraction Process
Weakness: The computational requirements for SWEb’s extraction model, which consumed 20,000 GPU hours on AMD MI250X GPUs, may be prohibitive for many research labs or developers working with limited resources. This factor could limit the pipeline’s accessibility and adoption.

Recommendation: Considering optimizations in the extraction model, such as fine-tuning on smaller, more frequent batches or experimenting with lighter transformer architectures, could reduce computational demands. Additionally, presenting a cost-benefit analysis comparing SWEb’s compute usage to the downstream performance gains would offer a more balanced view of the pipeline’s scalability and efficiency.

**Questions:**

N/A

---

> ### Author Response · Authors · 2024-11-20
>
> First and foremost, thank you for providing thorough and constructive feedback on our work. Your comments are well received and we'd like to address the pointed out weaknesses one-by-one:
>
> 1. Limited Applicability Beyond Scandinavian Languages Weakness: We acknowledge this point, as model based extraction is a generally applicable method. In future work we will take more language families into consideration.
>
> 2. Reliance on Manual Annotation in Model Training Weakness: Annotating thousands of samples for every new language could very well become a bottleneck. We would however expect the extractor model (with multilingual pre-training) to transfer quite well across languages, requiring less data for additional languages. With future work involving more languages (as pointed out in 1), experiments and learning curves investigating this would be very insightful. Great idea!
>
> 3. Evaluation Restricted to Swedish Benchmark Weakness: This is an important weakness you raise. As other benchmarks for these languages (e.g. Scandeval) didn't provide good signal at this small scale, we prioritized constructing an appropriate benchmark in the majority language (Swedish). Following up with corresponding tests in Danish, Norwegian and Icelandic is a top priority in subsequent work.
>
> 4 and 5. Limited Analyses of Extracted Content and Errors: We recognize further analysis is crucial to better understand and improve extraction from here. Recent work have shown content of specific kinds, e.g. educational, to be particularly effective towards improving downstream LM quality. We believe a targeted analysis particularly towards those "important documents" would be useful in improving the extraction further.
>
> 6. Computational Expense of the Extraction Process Weakness: The high costs with this model-based extraction is a limiting factor, despite 20k GPUh being small compared to the usage of actually training the LLMs. While we did explore training smaller extractors (not in the paper), with only small degradations in extractor performance, we chose to optimize for performance rather than efficiency in constructing SWEb. We will release training code and data for the extractor to allow for others to explore this direction further.

---

### Meta-Review · Area_Chair_d2MV · 2025-01-03

**Metareview:**

This paper introduces SWEb, a large-scale pretraining dataset consisting of over one trillion tokens for Scandinavian languages, presented with a detailed data collection and processing pipeline including a novel model-based text extractor. Reviewers recognized SWEb's significant contribution to the field, as it provides a high-quality resource for the traditionally under-resourced languages. The availability of this dataset, along with the new Swedish benchmark can allow the development of improved language models and significantly advance Scandinavian language research. Reviewers also appreciate the detailed methodology and open release of the resources.

One of the main concerns pointed out by reviewers is evaluation restricted to the Swedish benchmark. The authors mention that their study on other benchmarks for other Scandinavian languages (e.g. Scandeval) didn't provide a good signal at the small scale resulting in opting for the Swedish benchmark which is the majority language. A few of the other limitations of the work are the lack of qualitative and error analysis of the model-based extractor and its extracted content, the lack of ablation studies regarding markdown tags during pretraining, and the absence of an ethics section. The authors address these concerns in the rebuttal and mention adding them in the final version of the paper.

Given the substantial value of the dataset and the authors' efforts to address the reviewer’s concerns, I am recommending this paper to be accepted, with a suggestion to add the ethics section along with feedback mentioned by the reviewers in the final version of the paper.

**Additional Comments On Reviewer Discussion:**

See above.

---

### Decision · Program_Chairs · 2025-01-22

Accept (Poster)